# Risks of use and non-use of antibiotics in primary care: qualitative study of prescribers' views

Olga Boiko,[1] Caroline Burgess,[1] Robin Fox,[2] Mark Ashworth [ID] ,[1] Martin C Gulliford [ID] [1]

[1]School of Population Health and Environmental Sciences, King's College London, London, UK
[2]Bicester Health Centre, Bicester, UK

**Correspondence to**
Dr Martin C Gulliford;
martin.gulliford@kcl.ac.uk

## ABSTRACT

**Purpose** The emergence of antimicrobial resistance has led to increasing efforts to reduce unnecessary use of antibiotics in primary care, but potential hazards from bacterial infection continue to cause concern. This study investigated how primary care prescribers perceive risk and safety concerns associated with reduced antibiotic prescribing.

**Methods** Qualitative study using semistructured interviews conducted with primary care prescribers from 10 general practices in an urban area and a shire town in England. A thematic analysis was conducted.

**Results** Thirty participants were recruited, including twenty-three general practitioners, five nurses and two pharmacists. Three main themes were identified: risk assessment, balancing treatment risks and negotiating decisions and risks. Respondents indicated that their decisions were grounded in clinical risk assessment, but this was informed by different approaches to antibiotic use, with most leaning towards reduced prescribing. Prescribers' perceptions of risk included the consequences of both inappropriate prescribing and inappropriate withholding of antibiotics. Sepsis was viewed as the most concerning potential outcome of non-prescribing, leading to possible patient harm and potential litigation. Risks of antibiotic prescribing included antibiotic resistant and *Clostridium difficile* infections, as well as side effects, such as rashes, that might lead to possible mislabelling as antibiotic allergy. Prescribers elicited patient preferences for use or avoidance of antibiotics to inform management strategies, which included educational advice, advice on self-management including warning signs, use of delayed prescriptions and safety netting.

**Conclusions** Attitudes towards antibiotic prescribing are evolving, with reduced antibiotic prescribing now being approached more systematically. The safety trade-offs associated with either use or non-use of antibiotics present difficulties especially when prescribing decisions are inconsistent with patients' expectations.

## INTRODUCTION

Inappropriate (AB) antibiotic prescribing is widespread but may bring risks for individual[1 2] and population health from drug side effects as well as from growing antimicrobial resistance.[3] Conversely, antibiotic avoidance may be associated with risks from serious bacterial

## Strengths and limitations of this study

► The sample of participants was diverse, including different groups of primary care prescribers drawn from urban and rural settings.
► The views of respondents who participated in the study may not be representative of non-participating practitioners.
► Participant responses may have been influenced by the interview setting.
► Serious safety outcomes are infrequent and might not have been experienced by patients managed by participants in the study.
► The study may have limited transferability beyond high-income countries.

infections that could be avoided through earlier treatment of infection episodes.[2] Many studies have provided insights into the reasons for inappropriate antibiotic prescribing and several syntheses have been published,[4–6] but the safety gradient associated with reducing antibiotic prescribing has developed as a new and highly relevant area of research. In this paper, patient safety is understood as 'the avoidance, prevention and amelioration of adverse outcomes or injuries stemming from the process of healthcare'.[7] The risks associated with antibiotic prescribing decisions are a key element of patient safety and require in-depth analysis. This paper addresses the gap in knowledge about prescribers' perceptions of potential adverse outcomes associated with reduced antibiotic prescribing.

In the UK, primary care services account for nearly 80% of all medical antibiotic use but antibiotic utilisation in primary care has been declining in recent years and choice of antimicrobial agents has become more selective.[8 9] A national target proposes a further reduction in antimicrobial use of 15% by 2024[10] with antimicrobial resistance providing the rationale for the reduction in antibiotic prescribing. There were an estimated

60 788 antibiotic resistant infections in England in 2018[9] resulting from infection with diverse bacterial pathogens, additionally superinfection with *Clostridium difficile* may cause illness.[11] The scale of antimicrobial resistance is increasing, especially across middle-income and low-income countries.[12] Unnecessary exposure to antibiotics may also be associated with more immediate harms. As a result of prescribing in the community, antibiotic-associated adverse events including allergic reactions lead to many emergency visits with antibiotics accounting for up to 20% of hospital admissions from drug reactions in the USA.[13 14] On the other hand, withholding antibiotics might potentially carry risks and reduced antibiotic prescribing in general practice is associated with a small increase in complications such as treatable pneumonia and peritonsillar abscess.[2 15]

The perceived priority of risks from either prescribing or not prescribing antibiotics requires a nuanced explanation within the broader realm of professionals' perceptions of safety and associated risk management. Fear of the risk of bacterial complications[5 16] and prognostic uncertainty about potential outcomes when not prescribing[4 17] are reportedly among key factors that influence the prescription of antibiotics. Among hospital doctors, there is evidence that overtreatment is preferred to the potential for adverse patient outcomes from not prescribing.[18 19] Klein *et al*[20] and Broniatowski *et al*,[21] for example, demonstrate that medical decision-making tends to favour views that favour prescription ('why take risks') rather than on prescription avoidance ('antibiotics can be harmful'). In primary care, general practitioners (GPs) and other prescribers also deal with safety concerns in their decision-making, and a better understanding needs to be developed concerning the balance of risk between prescribing or non-prescribing of antibiotics.

Patient factors influencing decision-making on antibiotic prescribing include compliance with patient expectations and pressures.[16 22–24] Reducing AB prescribing in primary care is therefore highly dependent on successful management of patient expectations[25–27] and on shared decision-making.[28–31] It is known that clinicians weigh individual best practice against perceived patient satisfaction so that complex trade-offs are enacted.[32] Therefore, of research interest is also how the issues of safety and risk information are communicated to patients.

In the present study, we investigate how primary care prescribers perceive risk and safety concerns associated with reduced antibiotic prescribing.

## METHOD
### Study design
Semistructured interviews were conducted with primary care prescribers including GPs, nurses and pharmacists in two English regions, one an urban metropolitan area and the other shire town in England with a high demand for primary care services. The study was approved by London Hampstead Research Ethics Committee 18/

---

**Box 1  Interview guide**

What are the indications for antibiotic (AB) treatment?

To what extent do NICE (National Institute for Health and Care Excellence) (or local) guidelines influence your AB prescribing?

What are the risks of AB prescribing and non-prescribing?

How do you differentiate between infections and patients?

What are the common myths or stereotypes about antibiotics?

Can you give me an example illustrating the inaccurate understanding of their purpose, mechanisms of action, risks and consequences?

In your view, is there the best way to elicit and manage patient expectations regarding antibiotics?

How would you communicate the risks associated with both prescribing and non-prescribing antibiotics?

How confident are you in decision-making around AB prescribing?

Would you assess your approach to AB prescribing as always adequate and if so, what makes you think that?

Could you describe consequences of inappropriate treatment for infections?

What would be/were your actions following unresolved or repeated infections?

What is your understanding of antimicrobial resistance?

What are your goals and priorities in infection management?

Are there any social norms or group pressures that affect your professional practice with regards to AB prescribing and how?

Has your prescribing practice for antibiotics changed over the recent years?

Do you think patient expectations of AB treatment have changed over the recent years?

Are you aware of the prescribing practice of other HCPs (your colleagues) in relation to antibiotics? Have you ever had to challenge their prescribing decisions?

Has anyone challenged your own decisions?

How hopeful are you usually that the AB treatment is the best course of action?

Is it possible to assess both the short-term and long-term impact of AB treatment on the patients?

What is your decision-making strategy?

How anxious do you feel about the uncertainty around prescribing?

Which resources do you use to support your decisions on AB prescribing?

---

LO/1874 and participants gave written informed consent to participation.

### Interviews
An interview guide was developed (box 1), this was designed to address key elements of the substantive research topic; it was also loosely informed by elements of the Theoretical Domains Framework, which draws on behaviour change theory to understand factors influencing healthcare practice.[33–35] The interview guide was piloted with three GPs to ensure that the questions were appropriate, understandable and covered relevant prescribing behaviours. All interviews were conducted by the first author to ensure consistent quality. The interviewer has a PhD in medical sociology and is an experienced qualitative researcher. All interviews apart from one telephone interview were conducted face to face on general practice (n=26) and University (n=4) premises

in the period January to July 2019. The participants were offered £60 to acknowledge their contribution.

## Recruitment of participants

Metropolitan practices were invited to the study by the local Clinical Research Network who generated the expression of interest. A shire town high-demand practice was recruited through informal Clinical Research Network contact that also helped in liaising with potential respondents. Potential participants were then approached either directly via email using the study information pack or indirectly via the practice manager or lead GP. The information pack included the invitation letter and study information sheet. A reminder was sent out 2 weeks after the initial approach to those who had not responded. A purposive sampling approach was followed: all participants were prescribers. Forty-nine primary care prescribers from ten GP practices were invited and thirty agreed to take part. The sample size was determined using the pragmatic concept of 'information power',[36] taking into account the aim of the study, sample specificity, quality of dialogue and analysis strategy. The uptake varied between practices (in five practices only a single participant was interviewed).

## Analysis

The interviews were digitally recorded, transcribed by a professional transcriber, imported to an NVivo-12 project and coded through an iterative six phased process described in thematic analysis.[37] Data analysis occurred iteratively and involved familiarisation, coding, theme searching, theme reviewing, theme defining and naming and producing the report. Repeated patterns in the data formed the basis for the codes, identified by the first author, and one single code for every different concept/idea was generated. To ensure that codes were applied consistently, a co-author (CB) independently coded a random sample of four interview transcripts. Coding was refined after discussion. Data identified by the same code were collated together and all different codes were sorted into potential subthemes and themes using NVivo options of tree building. Then, the potential themes were reassessed and reorganised to reflect major narratives and themes in the coded data. Finally, the first, second and the last authors refined and named the themes and subthemes.

## Patient and public involvement (PPI)

Participants' feedback on the transcripts or the summarised final findings was not sought; however, the process of developing subthemes and themes was discussed at a PPI meeting. The purpose of the meeting was to inform the research of patient and service user perspectives. The meeting was attended by six PPI members including four women and two men of diverse ages. The preliminary findings were presented, and members were invited to discuss emerging themes and to review selected quotes from the interview transcripts for relevance. Feedback

**Table 1** Characteristics of participants. Figures are frequencies

| Characteristic | Variable | Number |
|---|---|---|
| Gender | Male | 8 |
| | Female | 22 |
| Location | Metropolitan | 21 |
| | Shire town | 9 |
| Occupation | GPs | 23 |
| | Nurse prescriber | 5 |
| | Pharmacist | 2 |
| Years of practice | <10 | 16 |
| | 10–20 | 10 |
| | >20 | 4 |

GPs, general practitioners.

included comments on patient expectations, patient pressure for antibiotics, trust and communication with GPs leading to additional interpretation.

## RESULTS

We recruited 30 participants from 10 general practices (table 1. Characteristics of the participants), including 23 GPs, 5 nurses and 2 pharmacists. The interviews lasted between 24 and 46 min. GPs', nurses' and pharmacists' responses were analysed as a single group because of the many commonalities and smaller number of non-medical respondents. We found there were no discernible differences in participants' accounts between the shire town and metropolitan settings. Overall, three participants expressed an overt avoidance of antibiotics, three others acknowledged overprescribing, while most prescribers leaned towards reduced prescribing. We distinguished three major themes from the data: risk assessment, balancing treatment risks and negotiating decisions and risks (table 2).

### Theme 1: risk assessment
#### Identifying treatment thresholds

The primary focus of diagnostic decision-making for participants was concerned with identifying major indications for antibiotic treatment. These were judged

**Table 2** Summary of main themes and subthemes

| Theme | Subthemes |
|---|---|
| Theme 1: Risk assessment | Identifying treatment thresholds |
| | Confidence in prescribing |
| Theme 2: Balancing treatment risks | Risks of prescribing and non-prescribing |
| | Facing antimicrobial resistance |
| Theme 3: Negotiating decisions and risks | Managing patient expectations |
| | Communicating risks |

to include the nature and severity of illness based on presentation of symptoms and signs, in the context of the patient's medical history. A majority of participants adopted a risk stratification approach in undertaking clinical assessment.

> It's a combination of things… For example, for an upper respiratory tract infection, tonsillitis, pharyngitis, you know, there's a Centor guidance. So that's where you have a checklist of things. Does this person have cervical lymphadenopathy? Do they have a fever? Do they have like absence of a cough, you know. Do they have exudate on their tonsil? So, then if you have a score of 3 or more then they have antibiotics. (Int 1, GP).

Risk stratification approaches included additional patient factors such as patient age and the presence of comorbidities including COPD, asthma, diabetes, cancer or a history of pneumonia. Whereas, many followed risk assessment protocols based explicitly on local or national clinical guidelines, some participants stressed the importance of clinical judgement in making safety-driven decisions.

> You don't want to miss something very serious. So, that's where your clinical judgement and decision-making skills play a major role. And experience, obviously, because these are things I deal with every day. (Int 14, Nurse)

Threshold-guided decision-making spanned the continuum from 'I am prescribing' to 'I am not prescribing'. Diagnostic uncertainty was part and parcel of the threshold-guided decision-making: prescribers pointed to the difference between more and less obvious cases, characterised by equivocal, ambiguous and non-convincing evidence. One participant contrasted several hypothetical scenarios:

> '…a patient with COPD, bronchiectasis, I may have a lower threshold for treating than a very fit and well 20-year-old, even if that 20-year-old had a productive cough with green sputum, their chest is clear, I'm not likely to give them antibiotics. Well they're not feverish, whereas if they're an 80 something with a history of COPD then I'd have a lower threshold for starting antibiotics because they're likely to have less reserve and more likely to have complications from an infection' (Int 20, GP).

### Confidence in prescribing

Appropriate prescribing and not just a reduction in antibiotics emerged as a priority for participants, who reflected on their own performance from different perspectives. In general, participants reported a high level of confidence in prescribing but also noted occasional limitations:

> I feel confident but that doesn't mean necessarily that I think I'm making the right decision in every

case. Sometimes when I'm making perhaps the wrong decision, I'm making that maybe because of patient pressure or because of my unwillingness to tolerate risk. (Int 22, GP).

Many participants acknowledged changes towards less prescribing over the last few years:

> I prescribe less because I guess we're more aware now of drug resistance than we were 5 years ago. It's much more talked about and we're seeing it more. But also, I'm now more confident in having that difficult discussion with the patient. (Int 5, GP)

### Theme 2: balancing treatment risks
#### Risks of prescribing and non-prescribing

Seven participants explicitly identified safety as a priority in infection management. All participants demonstrated vigilance to risks arising both from prescribing antibiotics and not prescribing. The fear was expressed of 'missing something' that could cause deterioration and consequently, participants admitted 'being cautious' and favoured prescribing antibiotics. At the same time, the common concern was also the avoidance of prescribing unnecessarily. Among the risks of prescribing, several side effects were reported, most commonly, gastrointestinal upsets, nausea, *Clostridium difficile* infection and thrush but also allergic, anaphylactic reactions, antibiotic resistance and less common side effects such as liver problems (failure). Participants also observed long-term adverse consequences of inappropriate prescribing:

> I think, certainly for children, I think if you prescribe antibiotics and they don't need them and then they have a rash because they've got a virus and then a penicillin allergy on their notes for the rest of their lives… I think another consequence is that if you prescribe inappropriately, it's very difficult for another healthcare professional, down the line, to explain to that patient, you're almost saying the other person was wrong. (Int 15, Nurse)

Risks of non-prescribing generated a shorter list with sepsis being the most concerning consequence.

> Sepsis…that's one thing I do worry about. If I see someone who's got a high temperature and a high heart rate… then I think about those factors and I think actually if this was in my clinical judgement – if I left this for 2 days, then I think they would be crossing that line (Int 26, GP).

Three prescribers who acknowledged the tendency to overprescribe, did so, in one case, because they assessed antibiotics' benefits to exceed harms and in two cases because of potential litigation following a missed serious bacterial infection:

> Because medico-legally you're much more likely to be brought up on missing something and not prescribing antibiotics than giving antibiotics when it wasn't

necessary… if there's any uncertainty about prescribing antibiotics I would always err on the side of giving them because the risk, however small of missing an infection that then gets worse would be enough for me to give antibiotics. (Int 19, GP)

### Facing antimicrobial resistance

Participants shared concern for the global rise in antimicrobial resistance. At the same time, they acknowledged lacking in-depth microbiological knowledge: "we talk more about not prescribing and prescribing correctly than resistance itself' (Int 9, Pharmacist). Meanwhile, they had to deal with the consequences of the antimicrobial resistance in their daily practice:

I've had a few patients that have had MRSA [Methicillin-resistant Staphylococcus aureus]. I've had a few people who have had PVL [Panton-Valentine leukocidin form of MRSA] infections, skin infections with multiple resistance… So we can sometimes struggle to find an antibiotic that's oral, that's then suitable. I've got a Type 1 diabetic, young lady, who has very poorly controlled diabetes and recurrent boils and abscesses on her back. And we did a swab of that and yes, there was only one oral antibiotic that was sensitive - everything else was resistant. (Int 23, GP)

antimicrobial resistance was most commonly encountered in older women with urinary tract infections:

'I think sometimes you do see, for example, in the UTI breakdown, some people have quite resistant UTIs and that becomes difficult'. (Int 15, Nurse)

'I've been a GP for about 10 years and you've already seen that certain antibiotics just aren't working anymore, and we need to change the way that we're doing things and you know we used to give trimethoprim locally first line for UTIs. Resistant in the majority of cases. So, we're giving nitrofurantoin'. (Int 10, GP)

There was mention of difficulties in conveying information about resistance to patients—discussing it in the encounters and emphasising community impact may have been less efficient than focussing on individual risks. There was also a worry that primary care is running out of antibiotics despite the strategies of second-line and third-line antibiotics:

They (patients) literally cannot have any, they've got an E. coli infection that's not sensitive to amoxicillin or nitrofurantoin or trimethoprim or even cefalexin or the cipro. It's just like literally multiply-resistant. And there's some quite virulent, my understanding is it is strains of bacteria where antibiotics will not work. And then you kind of get to the hard-core ones. (Int 11, GP)

In such cases of failure of several course of antibiotics, referral to secondary care, possibly for intravenous

therapy were reported as the only options. Other times, where the resistant organism could be tackled in primary care, the last resort was a longer course or long-term prophylactic antibiotics. More investigations and consultations with microbiologists about unresolved infections appeared to precede these decisions.

### Theme 3: negotiating decisions and risks
#### Managing patient expectations

Participants identified patient pressure as a factor in their decision-making but they shared the view that patients differ in terms of their expectations regarding antibiotics. On the one hand, increased knowledge of the appropriate indications for antibiotic therapy (not for viruses) and understanding of antimicrobial resistance from public health and media campaigns was noted. On the other hand, patient pressure in a form of implicit expectations or explicit demands remained frequent: readily prescribed in the past, antibiotics had a profile of immediate cure in large parts of patient population:

… so many people have been mis-prescribed antibiotics in the past that I think they just won't believe you that they don't need them (Int 25).

A GP summarised this ambivalence:

There's a reasonable cohort now who come in and say they don't want them (antibiotics). They've read, they're educated, they know that they're contributing potentially to resistance and they don't want to risk the side effects. But there's also a large cohort still who come in and say, "My cough's gone to my chest, I need antibiotics. So, it's trying to often you know, get through those barriers and explain to them that their chest is clear." (Int 10, GP)

Eliciting expectations, educating patients and delayed prescription were the key strategies for managing patient expectations. Explaining assessment results and positive language were deemed important for the success of the consultation; several participants preferred the time-saving mode of giving out written information about the expected length of illness (for example, about the duration of sinusitis with and without antibiotics) and about side effects of antibiotics. Elicitation of expectations included asking patients: "What were you hoping for when you came in today?" (Int 26, GP). Delayed prescriptions were used by all but three of the participants interviewed. This was considered as a form of partnership, of shared decision-making between the clinician and the patient:

… that helps patients because at least psychologically they have got an antibiotic, but they know they can't use it straightaway. (Int 25, GP)

#### Communicating risks

As above, participants demonstrated that the commitment to reduced prescribing was dependent on patient understanding of the need for antibiotics. This meant

that at times building and maintaining relationships were prioritised and led to prescribing decisions, as an interviewee reported:

> Much of my job is trying to build a rapport with someone and build a rapport so that we can have a conversation that's therapeutic. If someone has come in adamant that they want antibiotics there is some conversation to be had there. Why did you get this idea from? What is it that you believed this would do? And what is your previous experience? Now, if they're not willing to go into that today, I may actually give them a short course of antibiotics with the understanding that we have another conversation. This is a way of building some trust (Int 29, GP).

The participants differed in terms of how they dealt with risk in encounters with patients: some were liberal prescribers who tended to avoid complaints and patient frustration, others preferred having difficult conversations on non-antibiotics' course of actions. Among liberal prescribers, there was the notion of offering antibiotics in order to be safe. In the case of non-prescribing, prescribers sometimes delved into lengthy explanations in order to secure patient adherence:

> When I'm explaining that there's no sign of bacterial infection and we don't want to give you antibiotics if we don't need to. Most people go, 'Oh yes, yes, no, of course not.' But some people might say, 'Oh, well, you know.' Then I will go into the reasons why, you know. 'Well actually you might get side effects, you know, it can make you, give you diarrhoea, it can give you thrush. And things can become resistant to it and it won't be helpful for you in the future. (Int 30, GP)

Advice on possible warning signs: 'safety netting' emerged as a dominant risk reduction strategy:

> I will give them (patients) an awful lot of safety-netting, and tell them what, 'If this doesn't get better, this is when you come back.' You know, or 'These are the signs of you getting worse,' or what they do if they are getting worse. (Int 16, GP)

## DISCUSSION
### Main findings in comparison with previous research
The study describes primary care prescribers' perceptions of safety and associated trade-offs in the context of reduced antibiotic prescribing. We identify three key themes with relevance to safety: risk assessment, balancing treatment risks, and negotiating decisions and risks. These accounts from primary care demonstrated variations in prescribers' approaches to decision-making behaviour, including perceptions of risks associated with prescribing or not prescribing antibiotics and in the communication of these decisions and risks to patients.

Decision-making for appropriate antibiotic prescribing was informed by safety considerations.

Guideline-concordant risk assessment was generally preferred to tacit clinical judgement based on informal heuristics in line with previous research[38] Confidence in prescribing can be contrasted with views that accentuate diagnostic uncertainty.[4 17] In complex or uncertain cases, resolution was usually in favour of antibiotic prescribing, but this was in the context of a secular shift to generally more restrictive antibiotic prescribing behaviour. The reduction imperative coexists with liberal prescribing, which was influenced by low tolerance of risks and patient pressures. This corresponds with extant literature that identifies the coexistence of different prescribing behaviours including antibiotic compromising, antibiotic delaying and antibiotic withholding.[24]

Safety trade-offs emerged from the respondents' perceptions of risk by lending support to recent qualitative research, which reported the complexity of balancing risks of antibiotic prescribing in hospitals.[39] In addition to anticipated benefits, respondents identified multiple risks associated with either prescribing or not prescribing antibiotics, so that the immediate and long-term adverse effects of prescribing, including antimicrobial resistance, were weighed against potential complications of non-prescribing such as sepsis. These untoward consequences rendered risk a double-edge sword. In the theory of social systems, such a conundrum can be described by the distinction risk/danger, rather than risk/safety because there is no absolute safety in prescribing decisions, hence the other side of risk remains danger not safety.[40 41] From Luhmann's[40] perspective, some distinctions are two-sided forms of 'second-order' observations, where one side is actualised at any given moment, but both sides may have equal relevance to the situation. Risk/danger represents such a form which exemplifies the contingency associated with seemingly binary choices, but which in itself represents actuality versus potentiality. According to this perspective, safety experts are 'first-order' observers who may not account for the mutuality of contingency because the other side is always present on the background. Boiko *et al*[41] applied this understanding to the analysis of clinical risks associated with anticoagulant prophylaxis, where risks of thrombosis were complemented by dangers of contraindications (eg, bleeding). In our situation of antibiotic prescribing, the 'risk' side is associated with prescribing potentially resulting in antimicrobial resistance and side effects, while the other side (danger) can be actualised if non-prescribing is chosen and can become the actual risk through complications such as sepsis. We found variation in how the prescribers perceived this duality, with the safety argument contributing in both directions: prescribing and non-prescribing. In other words, professionals' acting on 'doing something' were juxtaposed against 'doing no harm' concerns. The participants were able to distinguish between short-term (eg, side effects) and long-term (eg, antimicrobial resistance, effect on doctor–patient relationship) trade-offs of prescribing. Antimicrobial resistance was generally viewed as a standalone long-term adversity now being encountered in daily

practice; it is gaining in prominence in contrast to findings from the earlier qualitative studies[42 43] and now has a more personalised relevance and clinical significance than some recent reviews suggested.[44]

Respondents negotiated safety in dealing with patients by rendering medical decision-making more explicitly during consultations. Patient expectations were found to be changing and so were the strategies employed in managing them. There was an emerging consensus on strategies to reduce antibiotic prescribing including patient education, improved self-management advice and delayed prescribing, supported by patient-centred communication emphasised in the other literature too.[45] At the same time, our study showed communication was primarily centred on warning signs, and on maintaining a clinician–patient relationship, rather than on the discussion of risks and benefits with patients. This is consistent with previous findings that explicit analysis of trade-offs is most often undertaken by physicians alone rather than as part of a dialogue with patients.[46] More explicit risk communication might become a focus of the consultations for (bacterial) infections. Systematic review evidence suggests that shared decision-making reduces prescribing[47] and our study also found that both delayed prescribing[48–51] and safety-netting appeared as effective strategies of shared decision-making.

## Strengths and limitations

The study provided a coherent analysis of the views of primary care prescribers drawing on the responses of participants working in rural and urban settings and including a sample that was diverse with respect to professional training and years of experience. The size of the sample may not have been sufficient to distinguish differences in approach between groups with different professional training, but this could be explored further in future studies. The study may possibly have reduced transferability to other settings beyond UK primary care or beyond high-income countries. Participants were necessarily informed of the nature and purpose of the research, consequently both their participation in the interview and the interview responses might have been influenced by research participation. It is possible that respondents who were less inclined to reduce antibiotic prescribing might have been less prepared to participate. Interview responses might have been inclined to give what they perceived as 'socially acceptable' responses. We employed a thematic analysis because this enables a flexible investigation of a complex topic without drawing on pre-existing theory. In order to reduce the possibility of inconsistency, we employed a systematic, staged approach to analysis and a sample of transcripts was repeat coded by a second analyst. A patient group was involved in the research, but we acknowledge that patient involvement contribution must be managed carefully to avoid introducing bias. The thematic analysis was completed by experienced qualitative researchers using participant data; PPI input did not in this case lead to any modification of themes identified.

It might be argued that if the PPI group did not materially influence the eventual data presentation, then the information about PPI involvement could be removed from the paper. However, the funders, the journal and the authors remain committed to the importance of PPI and have retained the PPI statement. This paper should be read in conjunction with our companion study, which explored the views of patients as participants.[52]

## Implications for further research and practice

This study explored and characterised primary care prescribers' perceptions of safety issues and risk management strategies relevant to reduced antibiotic prescribing. The study offers insights into primary care prescribers' perceptions and as such it emphasises the safety perspective within the current debate on antibiotic prescribing and antimicrobial stewardship. The study identified dilemmas that are recognisable in the course of daily primary care practice and can form the basis for future improvement and antimicrobial stewardship programmes. Our research paves the way for a cross-sectional survey of risk perceptions. It highlights the need for further development of risk stratification and risk communication tools such as decision-making checklists and evidence-based support tools. It also stresses the need for adequate training on antimicrobial resistance and reducing of antibiotics (such as GRACE-INTRO and REDUCE).[53 54] Safety netting had a strong presence in the interviews, however, as such is under-researched and requires further exploration. Our findings support the argument[31] that prescribers need more time to discuss the benefit–harm trade-off within shared decision-making as this may help to reduce antibiotic prescribing in primary care.

## CONCLUSIONS

Attitudes towards antibiotic prescribing are changing and becoming more nuanced. There is growing confidence in the capacity to reduce the rate of prescribing and to manage patient expectations, which are themselves undergoing change. There is growing recognition that there may be safety trade-offs associated with antimicrobial stewardship and this is linked to concerns about sepsis and other serious bacterial infections. There is a need to develop better quantified estimates of risk that can inform clinical decision-making and 'safety netting' advice given to patients. This will require further development of risk stratification estimates, as well as communication tools that enable these to be used in practice. Improved management of risks and benefits will help to inform future antimicrobial stewardship efforts.

**Acknowledgements** We would like to thank all participants in this study and Patient and Public Involvement group members.

**Contributors** OB and MCG designed the study. CB advised on the development of the interview guide. MA and RF advised on the recruitment strategy and facilitated its implementation. OB conducted the analysis with support from CB. OB drafted the paper. All authors commented on and approved the paper.

**Funding** The study was funded by the National Institute for Health Research (NIHR) Health Services and Delivery Programme (16/116/46). MCG was supported by the NIHR Biomedical Research Centre at Guy's and St Thomas' Hospitals. The views expressed are those of the authors and not necessarily those of the NHS, the NIHR, or the Department of Health. The funder of the study had no role in study design, data collection, data analysis, data interpretation or writing of the report. The authors had full access to all the data in the study and all authors shared final responsibility for the decision to submit for publication.

**Competing interests** None declared.

**Patient and public involvement** Patients and/or the public were involved in the design, or conduct, or reporting, or dissemination plans of this research. Refer to the Methods section for further details.

**Patient consent for publication** Obtained.

**Provenance and peer review** Not commissioned; externally peer reviewed.

**Data availability statement** The data that support the findings of this study are available from the corresponding author upon reasonable request.

**ORCID iDs**
Mark Ashworth http://orcid.org/0000-0001-6514-9904
Martin C Gulliford http://orcid.org/0000-0003-1898-9075

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
