## [Reviewer comments · BMJ Open]

ARTICLE DETAILS

TITLE (PROVISIONAL)	Risks of use and non-use of antibiotics in primary care. Qualitative study of prescribers' views
AUTHORS	Boiko, Olga; Burgess, Caroline; Fox, Robin; Ashworth, Mark; Gulliford, Martin C

VERSION 1 – REVIEW

REVIEWER	Eva Krockow University of Leicester, UK
REVIEW RETURNED	05-May-2020

GENERAL COMMENTS	This is a well-written qualitative research paper on the use of antibiotics in primary care. It identifies three major themes in the process of prescribers' risk evaluation. Not all the results are novel, but overall the article presents updated insights on the balancing of risks and benefits of antibiotic prescribing in the context of a changing patient and prescriber pool, who are increasingly aware of the risks associated with antibiotic overuse. I would recommend publication, but I have a few minor suggestions for improvement. 1. On page 9, please provide more information about the Patient and Public Involvement meeting (e.g. number of participants, meeting programme).2. On page 11, under the heading "confidence in prescribing", three general prescriber approaches/attitudes towards antibiotics are prescribed. These attitudes do not seem to reflect the theme of confidence, but rather point to general prescribing types or biases. It might be worth exploring these in a separate paragraph (possibly in the subsection of "balancing treatment risks").3. Please be consistent in using "AMR" as abbreviation for antimicrobial resistance (or don't use the acronym at all).4. On page 14, the first quote is not legible due to a formatting error. Please correct this.5. On page 18, it would be helpful to include more detail on the theory of social systems. Not all BMJ Open readers will be familiar with this and it provides an interesting theoretical framework that would benefit from further exploration.
---

REVIEWER	Gloria Cordoba University of Copenhagen, Denmark
REVIEW RETURNED	15-Jun-2020

GENERAL COMMENTS	This is a qualitative study reporting the results of 30 semi-structured interviews to investigate how primary care prescribers perceive risk and safety concerns associated with reduced antibiotic prescribing. The topic and the results are important to design effective antimicrobial stewardship campaigns. The study has been well designed and the paper is well written. However, it requires some minor improvement to be ready for publication:  1. I would suggest to replace the phrasing of the objective in the abstract for the one in the main text (lines 55-54). 2. Page 6 Line 54: perceived or perceive? 3. Page 8 Line 54: please insert the initials of the co-author 4. Table 1 interview guide is not mentioned in the main text 5. Page 19 lines 13-22: The study.... (I would suggest to move this paragraph to IMPLICATIONS section 6. The strengths and limitations section needs a more thorough reflection of the methodology. For example, was it a good decision to use thematic analysis instead of other approach? Yes/no why..., Do you think the responses of the participants were influenced by the interviewer? Yes/No why.... 7. I would suggest a graph to summarize the main themes and sub-themes.
---

REVIEWER	Valerie Ness Glasgow Caledonian University, Cowcaddens Road, Glasgow. Scotland, United Kingdom
REVIEW RETURNED	15-Jun-2020

GENERAL COMMENTS	I thought this was a very interesting and well-written paper. I would suggest some minor revisions around the following areas:  - use of non-qualitative terms. In both the strengths and limitations sections of the paper the term "validity" is used. Firstly I am not sure how this "validity" can be determined regarding the sample (perhaps further explanation is needed within the text) and secondly perhaps it would be better to relate to aspects of trustworthiness instead of validity. I would also advise that "generalisability" be changed to "transferability" -again relating to trustworthiness and more akin with qualitative research. Pg 4 - perhaps mention other side-effects e.g. C Diff here too as it is mentioned later in the discussion. Pg 5 (line 54) - "the larger study" is mentioned but no further information is provided. If more information is relevant perhaps it could be included, otherwise I am not sure if this information is needed No information is provided regarding the development of the interview questions -were these based on a literature review, a theoretical model, expert opinion etc? Could this be provided? Pg 8 - a breakdown in the numbers/types of participants is given in the abstract -could this be repeated here? Pg 9 - I am unsure why the development of the themes would be discussed at a PPI meeting. Does this not introduce bias? Surely these themes were developed by the 2 researchers, therefore what was the purpose of the PPI group involvement at this stage? I am not sure if this can be done due to the small numbers but it would have been interesting to have explored the differences between the different groups of prescribers. However, in any case I am not sure you can say that they were grouped together due to commonalities as there are also differences between these professional groups.
--

	Pg 13 - formatting issue -part of the quote seemed to be missing Pg 14 - 3.2. Int 29 needs a professional grouping beside their quote
--	--

VERSION 1 – AUTHOR RESPONSE

Reviewer: 1

This is a well-written qualitative research paper on the use of antibiotics in primary care.

Thank you for this feedback.

1. *On page 9, please provide more information about the Patient and Public Involvement meeting (e.g. number of participants, meeting programme).*

Thank you, we now add (page 7): ‘The purpose of the meeting was to inform the research of patient and service user perspectives. The meeting was attended by six PPI members including four women and two men of diverse ages. The preliminary findings were presented and members were invited to discuss emerging themes and to review selected quotes from the interview transcripts for relevance.’

2. *On page 11, under the heading “confidence in prescribing”, three general prescriber approaches/attitudes towards antibiotics are prescribed. These attitudes do not seem to reflect the theme of confidence, but rather point to general prescribing types or biases. It might be worth exploring these in a separate paragraph (possibly in the subsection of “balancing treatment risks”).*

Thank you, we agree and have transferred this sentence to the introductory paragraph, so that the insight is not lost (page 8).

3. *Please be consistent in using “AMR” as abbreviation for antimicrobial resistance (or don’t use the acronym at all).*

Thank you, we now refer to ‘antimicrobial resistance’ throughout.

4. *On page 14, the first quote is not legible due to a formatting error. Please correct this.*

Thank you, this has now been corrected.

5. *On page 18, it would be helpful to include more detail on the theory of social systems. Not all BMJ Open readers will be familiar with this and it provides an interesting theoretical framework that would benefit from further exploration.*

Thank you for this comment. We now provide additional explanation of Luhmann’s approach (pages 16 to 17).

Reviewer: 2

The topic and the results are important to design effective antimicrobial stewardship campaigns. The study has been well designed and the paper is well written.

Thank you for this feedback.

1. *I would suggest to replace the phrasing of the objective in the abstract for the one in the main text (lines 55-54).*

Thank you, this change has been made.

2. *Page 6 Line 54: perceived or perceive?*
Thank you, this change has been made.
3. *Page 8 Line 54: please insert the initials of the co-author*
Thank you, this change has been made.
4. *Table 1 interview guide is not mentioned in the main text*
Thank you, this has been corrected.
5. *Page 19 lines 13-22: The study.... (I would suggest to move this paragraph to IMPLICATIONS section*
Thank you for this suggestion, which we have adopted, please see response to next point.
6. *The strengths and limitations section needs a more thorough reflection of the methodology. For example, was it a good decision to use thematic analysis instead of other approach? Yes/no why..., Do you think the responses of the participants were influenced by the interviewer? Yes/No why....*
Thank you, the section on strengths and limitations has been comprehensively revised (page 17).
7. *I would suggest a graph to summarize the main themes and sub-themes.*
Thank you for this suggestion, we now add a new Table 3 that summarises the main themes and sub-themes.

Reviewer: 3

I thought this was a very interesting and well-written paper.

Thank you for this feedback.

- use of non-qualitative terms. In both the strengths and limitations sections of the paper the term "validity" is used. Firstly I am not sure how this "validity" can be determined regarding the sample (perhaps further explanation is needed within the text) and secondly perhaps it would be better to relate to aspects of trustworthiness instead of validity. I would also advise that "generalisability" be changed to "transferability" -again relating to trustworthiness and more akin with qualitative research.

Thank you, these suggestions have been adopted in the revised strengths and limitations section.

Pg 4 - perhaps mention other side-effects e.g. C Diff here too as it is mentioned later in the discussion.

Thank you, this has now been added (Page 4).

/ cont ..

Pg 5 (line 54) - "the larger study" is mentioned but no further information is provided. If more information is relevant perhaps it could be included, otherwise I am not sure if this information is needed.

Thank you, this has now been omitted.

No information is provided regarding the development of the interview questions -were these based on a literature review, a theoretical model, expert opinion etc? Could this be provided?

Thank you, we now explain (page 6): 'An interview guide was developed (Table 1), this was designed to address key elements of the substantive research topic; it was also loosely informed by elements of the Theoretical Domains Framework, which draws on behaviour change theory to understand factors influencing health care practice. (34-36)'

Pg 8 - a breakdown in the numbers/types of participants is given in the abstract -could this be repeated here?

Thank you, this has now been added.

Pg 9 - I am unsure why the development of the themes would be discussed at a PPI meeting. Does this not introduce bias? Surely these themes were developed by the 2 researchers, therefore what was the purpose of the PPI group involvement at this stage?

Thank you, we have now revised the section on PPI in response to reviewer 1, and we now incorporate this point. We now explain (page 7): 'The purpose of the meeting was to inform the research of patient and service user perspectives. The meeting was attended by six PPI members including four women and two men of diverse ages. The preliminary findings were presented, and members were invited to discuss emerging themes and to review selected quotes from the interview transcripts for relevance. Feedback included comments on patient expectations, patient pressure for antibiotics, trust and communication with GPs leading to additional interpretation.'

I am not sure if this can be done due to the small numbers but it would have been interesting to have explored the differences between the different groups of prescribers. However, in any case I am not sure you can say that they were grouped together due to commonalities as there are also differences between these professional groups.

Thank you, we now acknowledge this as a limitation, where we say 'The size of the sample may not have been sufficient to distinguish differences in approach between groups with different professional training but this could be explored further in future studies.'

We also modify the account (page 8) to read 'because of the many commonalities and smaller number of non-medical respondents.'

Pg 13 - formatting issue -part of the quote seemed to be missing Pg 14 - 3.2. Int 29 needs a professional grouping beside their quote

Thank you, this has been corrected.

VERSION 2 – REVIEW

REVIEWER	Gloria Cordoba University of Copenhagen Denmark
REVIEW RETURNED	30-Jul-2020
GENERAL COMMENTS	Thanks, all my comments were addressed. The paper is ready for publication

REVIEWER	Valerie Ness Glasgow Caledonian University Scotland, United Kingdom.
REVIEW RETURNED	04-Aug-2020

GENERAL COMMENTS	Thank you for making these revisions which has improved the manuscript. I have two points I wish to make -one major (1) and one minor (2); 1. Although the involvement of the PPI group is now clearer I feel that this is not the purpose of PPI involvement and adds bias to the findings. The themes and illustrative quotes should be developed from a thematic analysis, completed by experienced researchers using participant data. The participants in this study are prescribers and although it is always crucial to have the patient perspective this should be for another study which has patients as the participants. PPI groups should not be used to influence findings with their perspectives - this means they are acting as study participants. I am therefore concerned that the themes have been influenced by PPI members and would need reassurance that this is not the case. 2. The strengths and limitations section reads a bit like a list. Remove the "However" from line 16 and suggest restructuring the section about interview studies by removing line 20 and then linking the limitations with what was done to improve rigor - hopefully this will improve the flow. Many thanks.
---

VERSION 2 – AUTHOR RESPONSE

Reviewer 3 comments:

Comment 1: Although the involvement of the PPI group is now clearer I feel that this is not the purpose of PPI involvement and adds bias to the findings. The participants in this study are prescribers and although it is always crucial to have the patient perspective this should be for another study which has patients as the participants. I am therefore concerned that the themes have been influenced by PPI members and would need reassurance that this is not the case

Response: We now incorporate this point under strengths and limitations, where we now say (p18): 'A patient group was involved in the research, but we acknowledge that patient involvement contribution must be managed carefully to avoid introducing bias. The thematic analysis was completed by experienced qualitative researchers using participant data; PPI input did not in this case lead to any modification of themes identified. This paper should be read in conjunction with our companion study, which explored the views of patients as participants.(52)'

Comment 2: The strengths and limitations section reads a bit like a list. Remove the "However" from line 16 and suggest restructuring the section about interview studies by removing line 20 and then linking the limitations with what was done to improve rigor -hopefully this will improve the flow.

Response: We have now adjusted the wording of this section, as suggested by the Reviewer so that it now reads: 'The study provided a coherent analysis of the views of primary care prescribers drawing on the responses of participants working in rural and urban settings and including a sample that was diverse with respect to professional training and years of experience. The size of the sample may not have been sufficient to distinguish differences in approach between groups with different professional training, but this could be explored further in future studies. The study may possibly have reduced

transferability to other settings beyond UK primary care or beyond high-income countries. Participants were necessarily informed of the nature and purpose of the research, consequently both their participation in the interview and the interview responses might have been influenced by research participation.'

We hope you agree that these changes meet the requirements of the Reviewer and have improved our paper. Thank you for considering the revised version for possible publication.

VERSION 3 – REVIEW

REVIEWER	Valerie Ness Glasgow Caledonian University Scotland, United Kingdom
REVIEW RETURNED	15-Sep-2020
GENERAL COMMENTS	My only point is again in relation to PPI -sorry. If the PPI group did not influence the findings (which they should not have done as mentioned in the study limitations) then I am not sure what they did do and/or what the purpose of PPI was in this study. Selecting quotes still seems to be involved in the analysis of the data. If the authors are sure that they were not influential in the findings then I would be inclined to remove the information about PPI involvement from the paper.

VERSION 3 – AUTHOR RESPONSE

Reviewer 3 comments: 'If the PPI group did not influence the findings (which they should not have done as mentioned in the study limitations) then I am not sure what they did do and/or what the purpose of PPI was in this study. Selecting quotes still seems to be involved in the analysis of the data. If the authors are sure that they were not influential in the findings then I would be inclined to remove the information about PPI involvement from the paper.'

We now debate the point in the Discussion (page 18), where we now say 'It might be argued that if the PPI group did not materially influence the eventual data presentation, then the information about PPI could be removed from the paper. However, the funders, the journal and the authors remain committed to the importance of patient and public involvement and have retained the PPI statement.'

We hope that this comment meets your requirements. We note that a PPI statement is encouraged in the BMJOpen instructions to authors.

Thank you for considering this revision.